# *Listeria monocytogenes* Response to Anaerobic Environments

**DOI:** 10.3390/pathogens9030210

**Published:** 2020-03-12

**Authors:** Brandy N. Roberts, Damayanti Chakravarty, J.C. Gardner, Steven C. Ricke, Janet R. Donaldson

**Affiliations:** 1Department of Biological Sciences, Mississippi State University, Mississippi State, MS 39762, USA; roberts@biology.msstate.edu; 2Cell and Molecular Biology, The University of Southern Mississippi, Hattiesburg, MS 39406, USA; damayanti.chakravarty@usm.edu (D.C.); jc.gardner@usm.edu (J.C.G.III); 3Center for Food Safety, Department of Food Science, University of Arkansas, Fayetteville, AR 72704, USA; sricke@uark.edu

**Keywords:** *Listeria monocytogenes*, anaerobiosis, bile, acidic, proteomics

## Abstract

*Listeria monocytogenes* is a Gram-positive facultative anaerobic bacterium that is responsible for the disease, listeriosis. It is particularly lethal in pregnant women, the fetus, the elderly and the immunocompromised. The pathogen survives and replicates over a wide range of temperatures (4 to 42 °C), pH, salt and oxygen concentrations. Because it can withstand various environments, *L. monocytogenes* is a major concern in food processing industries, especially in dairy products and ready-to-eat fruits, vegetables and deli meats. The environment in which the pathogen is exposed can influence the expression of virulence genes. For instance, studies have shown that variations in oxygen availability can impact resistance to stressors. Further investigation is needed to understand the essential genes required for the growth of *L. monocytogenes* in anaerobic conditions. Therefore, the purpose of this review is to highlight the data on *L. monocytogenes* under known environmental stresses in anaerobic environments and to focus on gaps in knowledge that may be advantageous to study in order to better understand the pathogenicity of the bacterium.

## 1. Introduction

*Listeria monocytogenes* was initially isolated in England in 1926 [1]. It was first named *Bacterium monocytogenes* after fatalities occurred in many rabbits due to increased monocytes in the blood. *Listeria monocytogenes* can survive a wide range of temperatures, including refrigeration, and is found in meats, spreads and soft cheeses, as well as raw milk and milk products. The bacterium has been isolated from food processing plants, animal feces and soil. It is inactivated by cooking and pasteurization; however, if proper sanitation procedures are not practiced, the risk of infection to the public through food processing can lead to illnesses. For instance, multistate outbreaks of listeriosis—such as that associated with Blue Bell ice cream in 2015—and the public health risk they pose highlight key aspects of food safety and heighten the need for more research to seek an understanding of *L. monocytogenes’* survival and methods to eliminate this pathogen in food sources. 

*Listeria monocytogenes* gained interest as a major human pathogen following epidemics in the 1970s [2,3]. Good manufacturing practices and food safety laws have helped reduce the incidence of listeriosis; however, approximately 1500 infections and 255 deaths still occur annually [4]. Those that are most susceptible to this pathogen are pregnant women, fetuses, the elderly and the immunocompromised. Genetic and molecular biology studies have provided significant insight into host–pathogen interactions [5]. A listeriosis infection can manifest as symptoms ranging from gastroenteritis to more serious sepsis or meningo-encephalitis, while exhibiting a particularly high fatality rate in individuals with compromised immune systems and resulting in increased abortions in pregnant women. The bacteria tend to target the liver and the spleen [6,7,8]. *Listeria monocytogenes* also has the ability to infect the brain or fetus by crossing the blood–brain or feto-placental barriers [9,10]. 

The survival of *L. monocytogenes* in the digestive tract of food animals and in environments near or on surfaces associated with food processing needs further investigation to gain more insight on how foodborne outbreaks involving this pathogen propagate. The economic devastation to the food industry, a lack of trust from consumers, and loss of life are three fundamental reasons why future research needs to target the conditions that sustain *L. monocytogenes*. The majority of established studies have highlighted the responses of *L. monocytogenes* in aerobic environments. However, *L. monocytogenes* is a facultative anaerobe and encounters changes in oxygen availability during food processing and throughout the gastrointestinal tract [11]. Oxygen concentration does not impact the growth rate of most strains of *L. monocytogenes* [12,13]. However, changes in oxygen availability do influence the response to stressors encountered within the host, including changes in pH, osmolarity, invasiveness and bile. 

Stressors are typically applied in the food processing environment in a hurdle mechanism. However, a phenomenon known as “stress hardening,” which refers to the increased strength of a pathogen to survive otherwise lethal factors, can occur. Some stresses to which *L. monocytogenes* might become resistant include differences in oxygen, temperature, pH and osmolarity. The more effective approach is a stepwise addition of hurdles to prevent the hardening of the pathogen’s resistance to these barriers. Therefore, the adaptation of a particular stress can protect *L. monocytogenes*. This response should be considered when new or current technologies that attempt to counteract food-borne illnesses are introduced or modified [14]. As cultivation under anaerobic conditions has been found to significantly influence the survival of this bacteria, this review attempts to highlight the defined metabolic pathways, physiological changes and pathogenesis under anaerobic conditions. 

## 2. Survival of *L. monocytogenes* under Anaerobic Conditions

### 2.1. Metabolic Pathways

*Listeria monocytogenes* can only utilize certain carbon sources for energy; glucose is the preferred source of energy for the bacterium. However, in the instance that glucose is not readily available, *L. monocytogenes* must seek alternative energy sources. In anaerobic environments, it has been observed that pentose and hexose both support the growth of *L. monocytogenes,* while maltose and lactose tend to be more supportive of aerobic growth. Glucosamine and N-acetylglucosamine both support growth in aerobic and anaerobic environments [15]. 

*Listeria monocytogenes* grown in glucose-defined media showed that lactate, acetate, formate, ethanol and carbon dioxide are generated from anaerobic cultivation [16]. An increased lactate yield occurred under anaerobic conditions, but 17% of the glucose carbon metabolized resulted in other end products, suggesting that *L. monocytogenes* are not strictly homofermentative. Other metabolites formed were formate, ethanol and carbon dioxide. Lactate production was higher in anaerobic compared to aerobic conditions. Lactate supplementation was shown to increase Listeriolysin O (LLO) production in an anaerobic environment [17]. 

A common food preservative and significant fermentation acid in the intestine is propionate [18]. Propionate supplementation under aerobic conditions caused decreased adherent growth and had no effect on planktonic growth. However, under anaerobic conditions, a higher pH led to a reduction in planktonic growth and caused an increase in adherent growth. Under anaerobic conditions, the cultures supplemented with propionate accumulated lower concentrations of ethanol [19]. Interestingly, propionate supplementation increased acetoin production under aerobic, but not anaerobic conditions. Additionally, propionate supplementation increased LLO production under anaerobic conditions, but not aerobic conditions [19]. However, acetoin, which is also a metabolite of pyruvate, had no effect on metabolite production in either anaerobic or aerobic conditions [17]. Pyruvate supplementation also caused a significant increase in LLO production in both aerobic and anaerobic environments. The supplementation of the tricarboxylic acid (TCA) cycle intermediates (i.e., acetate, citrate, succinate and fumarate) increased LLO expression in anaerobic conditions [17]. 

### 2.2. Adaption to Changes in pH

Among the first stressors encountered by *L. monocytogenes* after the ingestion of contaminated foods are the acidic, microaerophilic conditions within the stomach. This is where the acid tolerance response and glutamate decarboxylase system become activated. The acid tolerance response has been found to increase in intensity under anaerobic conditions [20]. Using simulated gastric fluid, *L. monocytogenes* was found to have an increase in resistance to pH 2.5 under anaerobic conditions, suggesting that oxygen limitation influences acid survival. Additionally, the glutamate decarboxylase system, which enables *L. monocytogenes* to survive in gastric fluid and other environments with low pH levels, was found to exhibit a similar level of expression under anaerobiosis as in acidic conditions, suggesting that the responses to the two are connected [21]. 

According to the study of Nilsson et al. [22], the ability to extend the shelf life of packaged fresh and ready-to-eat (RTE) foods is dependent on oxygen depletion. The working hypothesis in this previous study was that the *L. monocytogenes* strain EGD-e makes the physical changes necessary to move from aerobic to anaerobic conditions and thereby displays decreased lag times if moved to an abrupt environment of low oxygen. To test this hypothesis, the authors utilized multidimensional protein identification technology (MudPIT) to demonstrate that the abrupt transition to low-oxygen conditions supports a minimal lag time in alterations in pH. This fact has far-reaching implications for packaged fresh and RTE foods under low-oxygen conditions where alkaline adaptation exists [22]. In order to test whether oxidative phosphorylation declined, alkaline-adapted and non-adapted cells were exposed to carbonyl m-chlorophenyl hydrazine (CCCP), a chemical which inhibits oxidative phosphorylation. If alkaline-adapted *L. monocytogenes* are more dependent on substrate-level phosphorylation instead of oxidative phosphorylation, increased survival rates would be present upon exposure to CCCP. The findings showed that adding CCCP lowered growth rates, while growth still continued for the cells grown at pH 9.0. These findings were further supported by a decrease in the presence of acetolactate decarboxylase, which is the final enzyme and molecular indicator in the bacterium’s anaerobic growth process. Significantly, these findings also revealed that vacuum packaging, which is common in food preservation, promotes the ability of *L. monocytogenes* to reach dangerous levels at a rate much faster than non-alkaline-adapted cells. In addition, this study concluded that *Listeria* Class Ia RNR was most likely expressed as an effort to scavenge other cells for the minimal oxygen available [17]. These findings are important where food safety regulations are concerned, since *L. monocytogenes* is able to multiply in low-oxygen environments faster than other non-adapted cells. Additionally, the potential for *L. monocytogenes* to become adapted to alkaline substances should be considered when determining the risk of contamination. 

### 2.3. Bile Tolerance

The ability of *L. monocytogenes* to cause listeriosis depends upon its ability to resist certain barriers/stressors encountered within the digestive tract, such as bile [11]. *Listeria monocytogenes* encounter bile excreted by the gall bladder into the duodenum during digestion [23]. Bile is comprised of bile acids, cholesterol, phospholipids and bilirubin [24]. Bile acids are able to disrupt the cell membrane and the DNA. In order to colonize in the intestines, *L. monocytogenes* must be able to survive approximately 500 to 1000 ml of bile that is secreted into the small intestine from the gall bladder [24,25,26]. *Listeria monocytogenes* can grow extracellularly within the gallbladder, revealing that this bacterium is able to survive in high concentrations of bile [27]. Although bile tolerance mechanisms have been widely studied, most research does not mimic the anaerobic conditions of the gastrointestinal tract. Therefore, more research is needed on the response of *L. monocytogenes* to bile under anaerobic conditions. 

*Listeria monocytogenes* is divided into four genetic lineages, with at least 13 different serotypes identified. Of the four lineages, lineage I and II contain a majority of the strains associated with human cases [28]. In a previous study that analyzed various serovars of *L. monocytogenes*, it was concluded that increases in resistance to bile, especially at an acidic pH, occur under anaerobic conditions, but do so in a strain specific manner [29]. The survival of *L. monocytogenes* in bile, pH dependency on bile extract, GDCA and TDCA toxicity, anaerobic cultivation and acid cross-protection were all analyzed for ten strains representative of six serovars. The data indicated that there was a direct correlation between bile resistance and pathogenic potential. *Listeria monocytogenes* was also found to be able to continue to grow in bile anaerobically at physiologically relevant concentrations and also could survive in higher concentrations following an extended lag phase under anaerobic conditions [30]. These data indicated that the response of *L. monocytogenes* to oxygen may influence and correspond to disease outcome. 

An additional study aimed to determine whether bile survival and the proteome varied under anaerobic conditions and if this occurred in a strain dependent mechanism [31]. The proteome analysis demonstrated an increase in proteins related to the cell envelop under anaerobic conditions; furthermore, variants were observed which related to morphological differences. Interestingly, bile salt hydrolase activity varied between strains under aerobic and anaerobic conditions. The results suggested that oxygen availability does impact bile resistance and that this response is strain specific. 

Another study analyzed the proteomes expressed by two virulent strains (EGD-e and F2365) and an avirulent strain (HCC23) [32]. To simulate the conditions encountered within the gallbladder, these bacteria were exposed to bile salts at pH 7.5 under anaerobic conditions. The study found that viability in the presence of bile salts varied among strains. The proteins associated with the cell envelope and cellular processes were expressed differentially in all three strains. Internalin protein, a transmembrane efflux protein and lipoprotein increased. Interestingly, the only protein expressed in the osmotic stress-response category was cysteine synthase. The study also indicated that the stress-response and repair proteins were expressed differentially among the strains following exposure to bile salts [32]. 

These facts suggest that either a component in the bile may impact survival and/or that pathogenic strains have specific survival mechanisms. These data also revealed that more research is needed to decipher the transcriptome of multiple strains in response to stressors encountered under the physiologically relevant conditions and in vivo [29]. 

### 2.4. Heat Tolerance

Heat tolerance in *L. monocytogenes* occurs in a strain-specific manner. Additionally, exposure to other stressors/conditions prior to treatment with heat can also contribute to heat tolerance [33]. For instance, the impact of temperature and atmosphere were analyzed on growth kinetics; anaerobic conditions promoted better growth rates [34]. Another study also indicated that resistance to heat was highest under anaerobic conditions [35]. These data indicate that anaerobic conditions help improve heat tolerance.

*Listeria monocytogenes* are readily found in raw milk and many other refrigerated products. Therefore, it is of upmost importance that knowledge about destroying the bacteria through pasteurization is known. Pasteurization involves applying a high temperature ranging from 63 to 80 °C for 16 s to 30 min depending on the food type, to destroy any food-borne pathogens before consumption. To endure pasteurization, *Listeria* must be able to tolerate high temperatures. However, the Food and Drug Administration and the U.S. Department of Agriculture have published that *Listeria* cannot withstand even their minimal pasteurization at 63 °C for 30 min. So, why is it thought that *L. monocytogenes* can withstand pasteurization? It could be due to the fact that heat resistance can be altered by several circumstances, including time, temperature, sub-lethal heat shock, glucose starvation and hydrogen peroxide. Of the items previously listed, glucose starvation, hydrogen peroxide and sub-lethal heat shock all lead to the expression of heat shock proteins, which have been found to be an important factor in bacterial thermo-tolerance. Hydrogen peroxide and superoxide are also critical to the restoration of heat damaged bacteria. Knabel, Walker, Hartman and Mendonca [35] conducted a study to determine if a sub-lethal heat shock at 43 °C could increase the pathogen’s thermal resistance. The results indicated that *L. monocytogenes* could survive the low temperatures for long periods required for pasteurization. Incubation under anaerobic conditions increased thermal resistance when sealed in tubes, but not in solid media. The study indicated that molecular O_2_ was responsible for the recovery of severely heat-injured cells [35]. Doyle et al [36] were able to recover *L. monocytogenes* after the pasteurization processing of contaminated milk products, but Crawford [37] stated that any viable *L. monocytogenes* recovered from a food product that has previously been pasteurized is likely a result of contamination due to pasteurization inducing injury and reducing viability to the cell. Further research into *L. monocytogenes* is needed before a conclusion can be reached upon whether or not *L. monocytogenes* can survive pasteurization and the high temperatures associated with this process [35]. 

### 2.5. Cold Tolerance

*Listeria monocytogenes* is a psychrotroph, being able to withstand refrigeration to −1.5 °C [38,39]. Refrigeration is typically utilized as a means to increase shelf life of products, therefore making it necessary to understand how this organism is able to survive these conditions. *Listeria* withstands cold temperatures by maintaining cellular membrane fluidity, stabilization of ribosomal structure and synthesizing compatible solutes [40,41,42,43,44,45]. In low temperatures, the adaptation in *Listeria’s* membrane involves an alteration in the methyl end of fatty acids from iso to anteiso, and the shortening of the length of the fatty acid chain [40,46]. The survival of *Listeria* at low temperature is further strengthened by accumulation of osmolytes like glycine, betaine and carnitine [47,48]. 

In one study, *Listeria* was found to have improved growth at temperatures of 5 or 10 °C in an oxygen restricted environment. Furthermore, in anaerobic cultures at 19 °C, the duration of lag phase was significantly decreased in anaerobic cultures as compared to aerobic cultures [49]. Previous cultivation in higher temperatures increased the lag phase when at refrigeration temperatures [49]. In another study, cold shock proteins, which are known to be involved in protection in low temperatures, were down regulated under anaerobic conditions [50]. The role of cold shock proteins in survival under anaerobic conditions has not been analyzed, especially in regard to strain to strain differences. This is an area that needs to be the focus of further research.

### 2.6. Invasion and Intracellular Survival

*Listeria monocytogenes* can invade a variety of different eukaryotic cells. Upon entering the cell, the bacterium is surrounded by a mildly acidic vacuole. With the assistance of PI-PLC, PC-PLC, and LLO, *L. monocytogenes* destabilizes the vacuolar membrane leading to the evacuation from the vacuole. The adaptation of metabolism to glucose-1-phosphate is needed by *L. monocytogenes* once it has entered the cytosol [51]. Virulence factors of *L. monocytogenes* are manifested at various intervals of infection. These factors are primarily located at the bacterial surface or are released. ActA is responsible for cellular movement and transmission inside the host cell through the use of host actin extensions. This also propels the bacteria into neighboring cells. Systemic invasion mechanisms necessary for the survival and progression of the *L. monocytogenes* pathogen include secretory proteins. These substances are involved in nutrient absorption, cellular communication, detoxification and the conquest of competing microorganisms. During pathogenesis, *L. monocytogenes* must secrete the appropriate proteins for adhesion, invasion, multiplication and survival in the host organism. However, how anaerobiosis influences the expression of these important virulence factors is not understood. 

To further build on the findings that *L. monocytogenes* can survive harsh conditions, such as the human gut, an experiment performed by Andersen et al. [6] demonstrated that anaerobic environments enhanced the infectivity of *L. monocytogenes*. The experiment was performed to determine if cultivation in anaerobic conditions prior to being ingested affects infectivity. Caco-2 cell cultures were infected with *L. monocytogenes* grown under restricted oxygen conditions. The data indicated that anaerobic cultivation prior to invasion increased infectivity 100-fold. These findings were then validated in a guinea pig model. The animals infected with anaerobic cultures of *L. monocytogenes* resulted in a greater number of bacteria in the intestines compared to those infected in aerobic conditions. Additionally, *L. monocytogenes* were found in the feces at higher concentrations in animals infected with the oxygen-restricted doses. The concentration of *L. monocytogenes* decreased in fecal material, and eventually was undetectable in animals dosed with the bacteria grown under unrestrictive conditions. However, in the animals given oxygen-restricted cells, *L. monocytogenes* were present between one and three logs greater [6]. Thus, it can be concluded that oxygen-restriction enables *L. monocytogenes* to be a more virulent pathogen in vivo. 

A recent study in gerbils also identified that *L. monocytogenes* cultured under oxygen-restricted conditions increased invasiveness in vivo. Specifically, the livers of infected animals had a significant increase in bacterial loads and also exhibited areas of hepatocellular necrosis. This study also examined the changes in the microbial community of infected gerbils and found that the abundance of Bacteroidales and Clostridiales increased in animals provided anaerobic cultured *L. monocytogenes* [52]. Together with these previous studies, these data indicate that anaerobic cultivation prior to infections increases invasiveness in vivo. 

## 3. Conclusions

*Listeria monocytogenes* is successful in passing through three major barriers in the host: the intestines, the blood–brain barrier and the feto-placenta barrier. This ability is central to its physiology [53]. This review has examined how *L. monocytogenes* survives various conditions under anaerobiosis. Studying this effect will hopefully provide more information into the causes and virulence of this pathogen. Studies have also shown that temperature and oxygen influence the survival of *L. monocytogenes* in a low pH environment. Food safety is of paramount importance, as pasteurization is the only way to kill the pathogen. It has been determined that strictly anaerobic recovery may allow for the quick and accurate enumeration of injured, foodborne pathogens [35]. Therefore, the government has instituted strict guidelines regarding *L. monocytogenes*. Due to the combination of the low-oxygen tension food packaging and the potential for the pathogen to become adapted to low pH levels, further studies are needed in the area of nutrient limitation encountered during pasteurization. This is due to a phenomenon known as stress hardening, or a kind of tolerance adaptation that the pathogen develops when encountering heat shock during pasteurization. Additionally, more strains of the pathogen need to be screened to gain more insight into how oxygen is used to regulate the genes needed for survival under the processes of stress and stress hardening. More studies are also required in the area of DNA repair after exposure to stressors. A better understanding of this pathogen would improve our knowledge of how adaptation to anaerobiosis aids in pathogenesis. As more studies are conducted that provide a better understanding to earlier insights, we can hopefully begin to see fewer fatalities related to *L. monocytogenes.*

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
