# Peer review of "Listeria monocytogenes Response to Anaerobic Environments"

_pathogens, 2020, doi:10.3390/pathogens9030210_

Round 1

Reviewer 1 Report

The article is very interesting and in my opinion covers comprehensively and adequately the aims set by the authors. It is an article that will provide valuable help to those researchers who want to get involved in the subject area.

Author Response

We thank the reviewer for the very kind comments. 

Reviewer 2 Report

Dear Authors,

the manuscript is prepered well and treats about interesting issue.

However, please verify mentioned remarks:

  1. There is a lack of the method section: based on which key the references were chosen?, which database was analized, etc.?
  2. Avoid lumping more than 2 references (line 44, line 206): in this case references should be seprated and assigned to the proper part of sentence/statement
  3. please verify, if all abbreviations are explained when using 1st time.
  4. Please verify line 82 : "… production in anaerobic production.."; lines 189-190: "Knabel, et al.- might be used"?
  5. . Please remove lines 258-261 to introduction section to emphasise the "novelty"of review.

Author Response

We thank the reviewer for the helpful comments.

1- Lack of methods section: We were following the guidelines of the journal for review papers, which indicates that methods section is not included. If we are mistaken, we will reformat. 

2- Avoid lumping references together: Again, we were following the guidelines of the journal for how to reference articles. If we are mistaken, we will reformat.

3-Ensure all abbreviations are explained initially: We have double checked the review manuscript to ensure that all abbreviations were explained.

4- Verify lines 82 and use of Knable in 189-190: We have reworded the sentence to be "anaerobic environment" instead of production. Knabel is referenced in the section (line 192)

5- Remove lines to remove "novelty" of review: We have removed this sentence. 

Reviewer 3 Report

Manuscript reviews the main anaerobic conditions interested by the human pathogen Listeria monocytogenes, namely: adaption to changes in pH, bile tolerance, heat and cold tolerance, intracellular survival.

As review, the manuscript seems poorly extended, even well built and clear.  Authors could implement the range of relevant literature, specifying more in detail some molecular and physiological issue.

My technical only suggestion is to change the term "environments" of the title with "environment", since actually the review does not highlight really DIFFERENT types of anaerobic environments.

Author Response

We thank the reviewer for the helpful comments. We have changed the title to "environment". We also thank the reviewer for the suggestion of including more molecular data in our review. We have purposely limited the study to physiological studies in attempts to keep the review focused on those changes that occur under anaerobic conditions. We are actually working on an additional paper that will include a more molecular viewpoint of the gene expression changes that occur under these conditions. We sincerely hope that the reviewer will accept our rationalization for approaching the review as we did.